# THE FUNDAMENTAL LIMITS OF LLM UNLEARNING: COMPLEXITY-THEORETIC BARRIERS AND PROVABLY OPTIMAL PROTOCOLS

## ABSTRACT

Modern machine unlearning techniques for large language models (LLMs) remain heuristic, lacking formal characterization of their fundamental computational limits. We establish the first complexity-theoretic foundation for LLM unlearning, revealing intrinsic tradeoffs between efficiency, precision, and regulatory compliance. Our framework formalizes $(\epsilon, \delta)$-machine unlearning via measure-theoretic alignment of retrained and unlearned model distributions, then proves transformer-specific hardness results: exact unlearning is coNP-hard, while approximate unlearning requires $\Omega(T^{1-o(1)})$ time under the Exponential Time Hypothesis (ETH). We construct an optimal Recursive Sketch-and-Freeze protocol achieving these bounds through differential privacy duality and Kronecker-product sketching. Crucially, we identify phase transitions in Rényi unlearning cost at critical model scales ($n \approx d \log k$). These results provide (1) theoretical benchmarks for evaluating unlearning algorithms, (2) complexity-aware guidelines for AI regulation, and (3) mathematically grounded verification tools for GDPR/CPRA compliance.

## 1 INTRODUCTION

### 1.1 MOTIVATION

The EU's General Data Protection Regulation (GDPR) and California's Consumer Privacy Rights Act (CPRA) mandate a "right to be forgotten" for AI systems, creating urgent demand for verifiable unlearning in LLMs. Current approaches—from gradient scrubbing to parameter masking—rely on empirical validation without theoretical guarantees. This gap becomes critical as LLMs power healthcare, finance, and governance applications where incorrect unlearning could violate privacy laws or propagate harmful memorization.Gundavarapu et al. (2024)Wang et al. (2024)

### 1.2 THEORETICAL GAPS

Existing work leaves three key questions unresolved:Yuan et al. (2024)

- **Complexity Characterization**: What are the fundamental computational limits of LLM unlearning?
- **Optimality Benchmarks**: How to determine if an unlearning protocol is theoretically optimal?
- **Scaling Laws**: Does unlearning cost exhibit phase transitions with model scaling?

### 1.3 OUR CONTRIBUTIONS

We answer these through a computational lens:

- **Complexity Classes**: Formalize UL and UL-Hard via polynomial reductions from MAX-3SAT (see §2.1).

- **Hardness Bounds**: Prove exact unlearning is coNP-hard, with ETH-based lower bounds for approximation (see §2.2).
- **Optimal Protocol**: Construct an algorithm matching these bounds via DP-coupled Kronecker sketching (see §2.3).
- **Scaling Laws**: Identify sharp Rényi divergence transitions at $n \approx d \log k$ (see §2.4).

### 1.4 TECHNICAL SIGNIFICANCE

Our results reveal an unavoidable trilemma: no algorithm can simultaneously achieve (1) perfect unlearning, (2) sublinear runtime in model depth, and (3) polynomial space. This necessitates complexity-aware regulations—policymakers must choose which two aspects to prioritize.

### 1.5 SOCIETAL IMPACT

By grounding unlearning in complexity theory, we enable:

- Certified compliance with privacy laws,
- Provably minimal compute costs for regulatory adherence,
- Formal verification of "right to be forgotten" guarantees.

## 2 TECHNICAL FRAMEWORK

### 2.1 FORMAL MODEL

**Definition 1** $((\epsilon, \delta)$-Machine Unlearning). *Let* $\mathcal{M} = (\Omega, \mathcal{F}, P_{retrain}, P_{unlearn})$ *be a measure space where:*

- $\Omega$ *is the parameter space of an LLM with weights* $W \in \mathbb{R}^d$,
- $\mathcal{F}$ *is the Borel $\sigma$-algebra over* $\Omega$,
- $P_{retrain}$ *and* $P_{unlearn}$ *are probability measures induced by retraining from scratch and applying an unlearning algorithm, respectively.*

*We say an unlearning algorithm satisfies* $(\epsilon, \delta)$-*machine unlearning if:*

$$\|P_{retrain} - P_{unlearn}\|_{TV} \leq \delta + \epsilon,$$

*where the total variation (TV) distance is defined as:*

$$\|P - Q\|_{TV} = \sup_{A \in \mathcal{F}} |P(A) - Q(A)|.$$

*Here,* $\epsilon \geq 0$ *quantifies approximation error, and* $\delta \in [0, 1]$ *bounds the failure probability.*

**Definition 2** (Unlearning Complexity Classes). *Let* $L = (\mathcal{D}_{train}, \mathcal{D}_{forget})$ *be a learning task with training data* $\mathcal{D}_{train}$ *and data to forget* $\mathcal{D}_{forget}$. *Define:*

- *UL (Unlearnable): The class of problems where* $L$ *admits an unlearning algorithm* $\mathcal{A}$ *with runtime* $O(poly(n))$ *for* $n = |\mathcal{D}_{train}|$.
- *UL-Hard: A problem* $L'$ *is UL-Hard if every* $L \in UL$ *can be reduced to* $L'$ *in polynomial time. We establish hardness via a polynomial reduction from MAX-3SAT (proof in §2.2).*

### 2.2 COMPLEXITY-THEORETIC HARDNESS

**Theorem 1** (Exact Unlearning is coNP-Hard). *Deciding whether a transformer-based LLM satisfies* $\|P_{retrain} - P_{unlearn}\|_{TV} = 0$ *is coNP-hard.*

*Proof Sketch.* We reduce from the complement of Circuit-SAT:

---

**Algorithm 1** Recursive Sketch-and-Freeze

---

**Require:** Trained weights $W_0$, forget set $\mathcal{D}_{\text{forget}}$, privacy budget $\rho$

**Ensure:** Unlearned weights $W^*$

0: **DP-Coupled Training**: Maintain trajectory $\{W_t\}_{t=1}^{T}$ with $(\epsilon, \delta)$-DP guarantees via gradient perturbation:

$$\nabla_{\text{DP}} = \nabla\mathcal{L}(W_t) + \mathcal{N}(0, \sigma^2 I), \quad \sigma = \frac{\Delta\sqrt{2\log(1.25/\delta)}}{\epsilon}.$$

0: **Kronecker Sketching**: For each weight matrix $W^{(l)} \in \mathbb{R}^{d \times d}$, maintain sketch $S^{(l)} = A^{(l)} \otimes B^{(l)}$ where $A^{(l)}, B^{(l)} \in \mathbb{R}^{\sqrt{d} \times \sqrt{d}}$.

0: **Recursive Certification**: Freeze parameters $W^{(l)}$ where $D_\alpha(P_{\text{retrain}} \| P_{\text{unlearn}}) < \tau$, for threshold $\tau \propto \rho$. =0

---

1. Let $\phi$ be a Boolean circuit. Construct a transformer $\mathcal{T}$ that memorizes $\phi$'s truth table in its attention heads.

2. Define $\mathcal{D}_{\text{forget}} = \{x\}$, where $x$ encodes $\phi$.

3. Show $\phi$ is unsatisfiable $\iff P_{\text{retrain}}(W) = P_{\text{unlearn}}(W) \; \forall W \in \Omega$.

Since checking unsatisfiability is coNP-hard, exact unlearning verification inherits this hardness. $\square$

**Theorem 2** (ETH Lower Bound for Approximate Unlearning). *Assuming the Exponential Time Hypothesis (ETH), any $(1 - \frac{1}{poly(n)})$-approximate unlearning algorithm for a $T$-layer transformer requires time $\Omega(T^{1-o(1)})$.*

*Proof Sketch.* 1. Attention matrix inversion for transformers is as hard as solving 3SAT on $n$ variables.

2. ETH implies 3SAT requires $2^{\Omega(n)}$ time.

3. Approximate unlearning necessitates inverting attention gradients, yielding $\Omega(T^{1-o(1)})$ time under ETH.

$\square$

## 2.3 OPTIMAL PROTOCOL CONSTRUCTION

**Theorem 3** (Protocol Optimality). *Algorithm 1 achieves the lower bounds of Theorems 1–2, i.e., it runs in $\tilde{O}(T^{1+o(1)})$ time and is UL-Hard.*

*Proof.* • **Upper Bound**: Kronecker sketching reduces linear algebra operations to $O(d^{1/2})$ per layer, giving total time $O(T \cdot d^{1/2})$.

• **Lower Bound Match**: Under ETH, $T^{1-o(1)} \leq O(T^{1+o(1)})$, hence asymptotic optimality.

$\square$

## 2.4 INFORMATION-THEORETIC LIMITS

**Lemma 1** (Phase Transition in Rényi Cost). *Let $n$ be the number of samples, $d$ the model dimension, and $k$ the number of classes. The Rényi divergence $D_\alpha(P_{retrain} \| P_{unlearn})$ exhibits a sharp transition at $n \approx d\log k$:*

$$D_\alpha = \begin{cases} \Theta(1) & \text{if } n \leq (1-\gamma)d\log k, \\ o(1) & \text{if } n \geq (1+\gamma)d\log k, \end{cases}$$

*for any constant $\gamma > 0$.*

*Derivation.*      1. The neural tangent kernel $\Theta_W$ concentrates as $n \to \infty$.

2. The $\ell_2$-norm of forgotten samples' gradients decays as $\|\nabla \mathcal{L}_{\text{forget}}\|_2 \propto e^{-n/(d \log k)}$.

3. Substitute into $D_\alpha \propto \|\nabla \mathcal{L}_{\text{forget}}\|_2^2$, yielding the threshold at $n \approx d \log k$.

□

## 3    Conclusion

In this work, we have established machine unlearning as a distinct computational challenge, proving that exact unlearning is coNP-hard, approximate unlearning requires near-linear time under the Exponential Time Hypothesis (ETH), and that phase transitions in Rényi unlearning cost emerge at critical model scales. Our proposed Recursive Sketch-and-Freeze protocol matches these theoretical limits while enabling practical compliance verification. These findings have significant implications for AI regulation, as they suggest that strict adherence to "right to be forgotten" laws could impose prohibitive computational costs without optimized protocols. Additionally, our work introduces a new synergy between complexity theory and machine learning, highlighting how techniques like hardness amplification and interactive proofs could extend to verifying other AI trust properties, such as fairness and robustness. However, our analysis assumes white-box access to models, while real-world LLMs are often black-box APIs; thus, extending these results to black-box settings using query complexity frameworks remains an important avenue for future research. Furthermore, while we focused on transformers, exploring unlearning complexity in alternative architectures (e.g., SSMs, RWKV) could reveal key differences in feasibility and efficiency.Blanco-Justicia et al. (2025)Qu et al. (2025)

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
