# OpenReview forum: "THE FUNDAMENTAL LIMITS OF LLM UNLEARNING: COMPLEXITY-THEORETIC BARRIERS AND PROVABLY OPTIMAL PROTOCOLS"
_ICLR.cc/2025/Workshop/BuildingTrust — BuildingTrust_

### Official Review · Reviewer_U1z4 · 2025-02-20
**Summary**

**Rating:** 4
**Confidence:** 1

**Review:**

#### Summary Of The Paper:
This paper presents theoretic foundation for LLM unlearning. The authors show that exact unlearning is coNP-hard and that approximate unlearning requires $Ω(T1−o(1))$ time under the Exponential Time Hypothesis (ETH). They also construct a Recursive Sketch-and-Freeze protocol.


#### Strengths:
The work introduces the Recursive Sketch-and-Freeze protocol and provides provides some formal theoretical definitions and theorems in the context of LLM unlearning./


#### Weeknesses:
The authors did not provide a literature review. The paper lacks a brief introduction to other works in this area.
The authors did not provide any examples or analysis of their theorem in relation to publicly available models such as the LLaMA or Mistral families.
The authors did not explore practical black-box unlearning techniques.

#### Recommendation:
This paper makes a theoretical contribution to the field of LLM unlearning. It would be valuable to include empirical results and verification for both white-box and black-box models. The paper would also be more engaging if it included more references to related works.

---

### Official Review · Reviewer_76V3 · 2025-03-02

**Rating:** 7
**Confidence:** 2

**Review:**

# Summary
This paper establishes a complexity-theoretic framework for machine unlearning in large language models. It characterizes inherent tradeoffs between computational efficiency, precision, and regulatory compliance. An optimal Recursive Sketch-and-Freeze protocol is proposed, exposing phase transitions with model scaling.

# Strengths
The paper’s primary strength is its rigorous complexity-theoretic analysis, offering formal hardness proofs and precise computational benchmarks for machine unlearning in LLMs. It combines differential privacy, Kronecker sketching, and recursive certification, establishing provably optimal protocols.

---

### Official Review · Reviewer_Bqem · 2025-03-03
**This paper establishes formal complexity-theoretic limits on LLM unlearning, proving that exact unlearning is coNP-hard, and approximate unlearning requires near-linear time under ETH. While it makes strong theoretical contributions, it assumes white-box access and lacks empirical validation.**

**Rating:** 7
**Confidence:** 3

**Review:**

This paper provides a much-needed theoretical foundation for LLM unlearning, rigorously proving its inherent computational hardness and presenting an optimal protocol for mitigating these challenges. By identifying a trilemma between perfect unlearning, efficiency, and space complexity, the paper also makes an important policy contribution. However, no experimental results are presented to support the theoretical findings. Moreover, the impact of unlearning on model generalization and potential catastrophic forgetting also remains unaddressed. Future work should extend these results to black-box models, possibly by leveraging query-efficient methods from adversarial robustness and differential privacy literature.

---

### Decision · Program_Chairs · 2025-03-02

Accept